# Cu-Doped-ZnO Nanocrystals Induce Hepatocyte Autophagy by Oxidative Stress Pathway

**DOI:** 10.3390/nano11082081

**Published:** 2021-08-17

**Authors:** Qianyu Bai, Yeru Wang, Luoyan Duan, Xiaomu Xu, Yusheng Hu, Yue Yang, Lei Zhang, Zhaoping Liu, Huihui Bao, Tianlong Liu

**Affiliations:** 1College of Veterinary Medicine, China Agricultural University, No.2 West Road Yuanmingyuan, Beijing 100193, China; 15046900669@163.com (Q.B.); sy20193050805@cau.edu.cn (L.D.); xuxiaomu92@126.com (X.X.); hys_test@163.com (Y.H.); yangyue_1@cau.edu.cn (Y.Y.); 2Key Laboratory of Food Safety Risk Assessment, China National Center for Food Safety Risk Assessment, No.37 Guangqu Road, Chaoyang District, Beijing 100022, China; wangyeru@cfsa.net.cn (Y.W.); zhanglei@cfsa.net.cn (L.Z.); liuzhaoping@cfsa.net.cn (Z.L.)

**Keywords:** CZON, biodistribution, dose-dependent toxicity, autophagy

## Abstract

As a novel nanomaterial for cancer therapy and antibacterial agent, Cu-doped-ZnO nanocrystals (CZON) has aroused concern recently, but the toxicity of CZON has received little attention. Results of hematology analysis and blood biochemical assay showed that a 50 mg/kg dosage induced the increase in white blood cells count and that the concentration of alanine aminotransferase (ALT), superoxide dismutase (SOD), catalase (CAT), and Malonaldehyde (MDA) in the serum, liver, and lungs of the CZON group varied significantly from the control mice. Histopathological examinations results showed inflammation and congestion in the liver and lung after a single injection of CZON at 50 mg/kg. A transmission electron microscope (TEM) result manifested the autolysosome of hepatocyte of mice which received CZON at 50 mg/kg. The significant increase in LC3-II and decrease in p62 of hepatocyte in vivo could be seen in Western blot. These results indicated that CZON had the ability to induce autophagy of hepatocyte. The further researches of mechanism of autophagy revealed that CZON could produce hydroxyl radicals measured by erythrocyte sedimentation rate (ESR). The result of bio-distribution of CZON in vivo, investigated by ICP-OES, indicated that CZON mainly accumulated in the liver and two spleen organs. These results suggested that CZON can induce dose-dependent toxicity and autophagy by inducing oxidative stress in major organs. In summary, we investigated the acute toxicity and biological distribution after the intravenous administration of CZON. The results of body weight, histomorphology, hematology, and blood biochemical tests showed that CZON had a dose-dependent effect on the health of mice after a single injection. These results indicated that CZON could induce oxidative damage of the liver and lung by producing hydroxyl radicals at the higher dose.

## 1. Introduction

Nanomaterials are kinds of materials whose size are less than 100 nm in one dimension [1]. In two decades, safety issues and potential toxicity of nanomaterials (NM) has attracted growing attention across the world [2,3,4,5]. More and more emerging nanomaterials have been applied comprehensively in many fields, such as medicine, chemical engineering, and technical engineering [6,7]. Rapid development of nanotechnology faces a plethora of possible health and environmental challenges. In recent decades, although there have been extensive studies on the toxicity of nanomaterials (NM), present studies were often contradictory and fragmented [8,9]. Furthermore, the largest number of these previous researches have mainly focused on cell culture systems in vitro; however, the data drawn from these studies could be astray and require verification from experiments in vivo based on animals models [10,11,12]. Organism is extremely complex and so does the interactions of biological components with the nanostructures in vivo. As of right now, toxicology evaluations experiments in vitro cannot completely replace animal experiments of NMs [13,14,15]. For these reasons, the potential toxicological effect evaluation of NMs is a considerable puzzle, and more systemic researches are needed to illustrate that.

As a novel nanomaterial, CZON has made remarkable progress in the field of biological applications in recent years. CZON nanoparticles have been applied in antibacterial, antitumor, and other fields due to their prominent performance [16,17,18,19]. Especially in antibacterial fields, CZON have been the promising new antibacterial agents due to their insecure surface chemical properties [20,21]. Although CZON displayed great perspective in medical application, the lack of systemic toxicity research of CZON in vivo and in vitro was the major obstacle that held back further development in medicine [22]. There were few toxicity evaluations of CZON; however, these researches mainly focused on toxicity in vitro [23,24]. As mentioned above, data provided by previous studies in vitro may be misleading and need to be verified in animal studies. Furthermore, the interactions of biological components with CZON in vivo, such as the fate, kinetics, clearance, and metabolism of CZON in vivo, will be demanded urgently.

Autophagy can cause programmed cell death by degrading membrane-isolated cytoplasm and organelles [25]. In very recent years, autophagy caused by nanomaterials has intensified this concern [26,27,28,29,30]. Nanoparticle-induced autophagy is crucial for its metabolism, cytotoxicity, and therapy potency [31]. Jin et al. found that iron oxide could promote macrophage autophagy through activation of toll-like Receptor-4 signaling [32]. Zhou reported that 2D MoS2 nanosheets induced autophagy via perturbing cell surface receptors and mammalian target of rapamycin (mTOR) pathway from outside of cells [33]. As reported in the previous researches, reactive oxygen species (ROS) produced by nanomaterials was considered as an important impact factor in the autophagy formation [34,35,36]. Chou and his co-workers found that mesoporous silica nanoparticles induced oxidative stress, leading to high-level ROS release and autophagy-mediated necrotic cell death eventually [37]. Ma et al. reported that surface-modified gold nanospikes could induce the up-regulation of autophagy-related protein LC3-II in cancer cells involved in the increased ROS, mitochondrial depolarization, and cell cycle redistribution [38]. However, the role that autophagy plays in the toxicity of CZON is still undercover.

In the present study, autophagy caused by CZON was investigated based on the acute toxicity of CZON when they entered the blood stream by intravenous injection for one time. Malonaldehyde (MDA), superoxide dismutase (SOD), catalase (CAT), and glutathione (GSH) in the serum, liver and lungs were examined of mice received CZON in the study. TEM and western blot were used to detect the autophagy caused by CZON in the liver. Through EPR (electron paramagnetic resonance) examination, it was confirmed that oxidative stress played an important role in the toxicity induced by CZON. To better understand the fate of nanoparticles entered the body, we also investigated the biodistribution and clearance of CZON in different organs.

## 2. Materials and Methods

### 2.1. Fabrication and Characterization of CZON Nanocrystals 

CZON was prepared via a method, as previously described [39]. The obtained CZON was collected by centrifugation and washed three times with deionized water. Morphology and structure of the resulting CZON nanoparticles were observed with a JEOL-200CX transmission electron microscope (TEM, JEOL, Tokyo, Japan) The size of nanomaterial was measured by DLS (Zetasizer 3000HSA, Malvern, UK) using the suspension liquid of CZON. Raman spectroscopy was also used to investigate the structure of CZON (Raman DXR3, Thermo, MA, USA) by applying CZON suspension.

### 2.2. Animals Experiments 

Female CR mice, aged 6–8 weeks, were purchased from Weitonglihua experimental animal Co., Ltd. (Beijing, China) and used in the experiments. Five mice were divided into a group and housed in stainless steel cages with containing sterile paddy husk as bedding in ventilated animal rooms. The mice were acclimated in the controlled environment (temperature: 22 ± 1 °C; humidity: 60 ± 10% and light: 12 h light/dark cycle) with free access to water and a commercial laboratory complete food. The animal experiments were approved by the Animal Care and Use Committee of China Agricultural University (CAU) (permit number: 20140115-089). We followed the guidelines of the CAU Animal Care and Use Committee in handling the experimental animals during this study.

### 2.3. Single Dose Toxicity 

Forty mice bred by Institute of Cancer Research (ICR mice) were used in the dose probing study, including 200, 100, 50, and 25 mg/kg groups of CZON (Appendix A). In the single-dose toxicity study of CZON, a series of doses (10, 17.5, 24.5, 35 and 50 mg/kg) were set according to the pilot results. CZON was suspended in sterile 5% glucose (10 mg/kg) and administrated through the mouse tail vein. The mice received the same volume of sterile 5% glucose which was used as controls by intravenous injection. The body weights of mice were measured and recorded through the entire experiments. At 14 days after injection, all animals were sacrificed and samples were collected.

### 2.4. Coefficients of Major Organs, Hematology and Blood Biochemical Assay

The major organs, such as the liver, spleen, heart, kidneys, lungs, and brain, were excised and weighed accurately. The coefficients of these tissues to body weight were calculated as the ratio of tissues (wet weight, mg) to body weight (g). At 14 days post-administration, blood sampling was performed for hematology analysis using a standard saphenous vein blood collection technique. Standard hematology markers were selected for the analysis. For blood biochemical assay, the serum was separated after being centrifuged twice at 3000 rpm for 10 min from blood samples which were then collected via the ocular vein. Serum levels of alanine aminotransferase (ALT), aspartate aminotransferase (AST), blood urea nitrogen (BUN), and creatinine (Cr) were determined. These parameters were all assayed by using a Biochemical Auto analyzer (Type 7170, Hitachi, Tokyo, Japan).

### 2.5. Histopathological Examinations

Immediate necropsy was performed after death or sacrifice, and major organs recovered were fixed in 10% formalin. Then, histopathological examinations was performed using standard techniques. After staining, pathologist observed and analyzed these slides in a double-blind way using optical microscope (Olympus, X71, Tokyo, Japan).

### 2.6. Biochemical Analysis of Oxidative Stress

The serum was separated after being centrifuged twice at 3000 rpm for 10 min from blood samples. Endogenous antioxidants, such as superoxide dismutase (SOD), catalase (CAT), malonaldehyde (MDA), and reduced glutathione (GSH), were measured by test kits (Nanjing Jiancheng Bioengineering Institute, Nanjing, China). The procedure of these indexes were performed, as per the manufacturer’s instructions. For the liver and lung samples, homogenization of tissues were employed before the detection of SOD, CAT, MDA, and GSH.

### 2.7. TEM Imaging of Tissues 

At 24 h after injection, the liver tissues recovered from the mice received CZON at 24.5 mg/kg immediately fixed in 3% glutaraldehyde. Then, the samples were treated according to the general procedure for the TEM study. Briefly, the tissues embedded by resin and prepared the ultrathin sections (60 nm) by ultramicrotome were stained with lead citrate and uranyl acetate. The ultrathin sections were observed and analyzed on a Hitachi H-7650 TEM (Tokyo, Japan), operating at 80 kV by a pathologist in a double-blind way.

### 2.8. Western Blotting Analysis 

Western blot analysis was used to detect the protein expression levels of LC3II and p62 in the liver of mice which received CZON at 50 mg/kg. The LC3II and p62 antibody were obtained from Boster Biotechnolgoy Co., Ltd. (Wuhan, China). The detailed procedure of WB analysis was listed in the Appendix A.

### 2.9. Electron Paramagnetic Resonance Test of CZON

In order to detect the hydroxyl radical produced by CZON, as the trapping agent, DMPO was added into the solution, dispersed in an aqueous solution, and fully mixed for preparation. Capillaries loaded in the samples were applied to collect information about the radicals by an electron paramagnetic resonance machine (JEOL, JES-FA200, Tokyo, Japan).

### 2.10. Cu content Analysis

After being injected with the dose of 24.5 mg/kg CZON, three mice were sacrificed at the time point of 1 day, 7 days, and 28 days. Meanwhile, in order to detect Cu content, the ing. Finally, inductively coupled plasma-Optical Emission Spectrometer (ICP-OES, NexION 300X, Beijing, China) was applied to detect the Cu content in tissues with the guidance of manufacturer’s instructions.

### 2.11. StatisticsX

Results were displayed as mean ± standard deviation (S.D). A one-way analysis of variance (ANOVA) test using SPSS 14.0 (SPSS Inc., Chicago, IL, USA) was applied to carry out the multigroup comparisons of the means. *p* < 0.05 was regarded as the statistical significance for all tests of this study.

## 3. Results

### 3.1. Preparation and Characterization of CZON

Figure 1A showed that CZON had good monodispersion in deionized water and the size of CZON was about 10 nm. Figure 1B displayed that the average size of the nanoparticles was about 14 nm. X-ray Powder Diffraction (XRD, Rigaku SmartLab, Toyko, Japan) demonstrated that nanoparticle’s powder was the crystalline structure (Figure 1C). The doping molar ratio of Cu was 3%. Compared with the ZnO standard map of pure wurtzite, the structure of zinc oxide after Cu doping was a wurtzite structure, and the doping did not change the symmetry of the crystal structure. There was no copper peak formed after doping because the doped copper entered the zinc oxide structure. No other impurity peaks were observed, depicting the purity of the compound.

### 3.2. Single Dose Toxicity of CZON

In the dose-probing experiment, there were five and three mice died after being injected with a dose of 200 and 100 mg/kg CZON, respectively (Appendix A). However, no death could be observed when the dose of CZON was less than 100 mg/kg. Therefore, when it comes to the single dose toxicity research of CZON, mice were injected with 10, 17.5, 24.5, 35, and 50 mg/kg CZON, respectively. No death and unusual behaviors could be observed among all of the treated groups, while body weight was decreased by 10% after receiving CZON at 50 mg/kg (Appendix A). Besides, the coefficients of the liver, spleen, heart, kidney, lung, and brain were calculated after being sacrificed at 14 days. No obvious changes were observed in coefficients indexes of mice received CZON at all doses (Appendix A).

### 3.3. Hematology and Blood Biochemical Assay

Compared with the control group, there was no significant difference of all representative hematology indexes with the dose of 10, 17.5, 24.5, and 35 mg/kg (Figure 2). However, when it comes to the 50 mg/kg group, compared with the control group, there was no obvious difference of all indexes, except white blood cell counts (*p* < 0.05) which experienced an upward trend (Figure 2). As for blood biochemical assay, the ALT levels showed a dose-dependent manner after intravenous injection. Furthermore, the serum ALT levels showed an increased trend after being injected with 50 mg/kg (Figure 3A), while no significant changes were observed in other biochemical indexes among other groups (Figure 3B–D).

### 3.4. Pathological Changes in Mice

In live, no obvious changes in appearance and micro-morphology could be seen after CZON injections at 10, 17.5, 24.5, and 35 mg/kg (Figure 4A–C). However, CZON induced lymphocytic infiltration, microgranulation, and degenerative necrosis of hepatocytes at 50 mg/kg (Figure 4D). No obvious changes in the lung were observed in other dosages groups compared to the control group. (Figure 5A–C). Congestion and lymphocytic infiltration were observed in the lung of mice which received CZON at 50 mg/kg (Figure 5D). There were no significant changes for the spleen (Appendix A), kidney, and lung (Appendix A), according to the observation of morphology at all doses. The above results revealed that the acute toxicity of CZON mainly caused liver and lung damage when CZON was intravenously administrated. Furthermore, a dose-dependent manner was observed after single injection. More importantly, these results motivated us to do a further study about the mechanism of CZON-induced liver damage.

### 3.5. Indexes of Oxidative Stress

For serum samples, both SOD levels (Figure 6A) and the catalase activity (Figure 6B) experienced a downward trend in CZON-treated animals (at 35 and 50 mg/kg) compared to that of the control group (*p* < 0.05). However, the MDA content was much higher than that of the control group (*p* < 0.01) and for the 50 mg/kg group (Figure 6C). Compared with the control group, the above three indicators had no significant change in other dose groups. As depicted in Figure 6D, there was no obvious difference observed between the mice which received CZON and the control animals. As for the liver sample, Figure 7A,B shows the reduced levels of SOD and the catalase activity in the liver of mice which received CZON (50 mg/kg) compared to untreated control mice (*p* < 0.05), respectively. However, the MDA content of the CZON group (50 mg/kg) experienced an upward trend compared with the control group (*p* < 0.05) (Figure 7C). No significant changes of these above three indexes between the other dose groups and the control group were observed. Meanwhile, Figure 7D presents that there was no obvious change between the mice received which CZON and the control animals, in terms of GSH.

For the lung sample, the reduced levels of SOD in the lung of CZON-treated animals (50 mg/kg) are displayed in Figure 8A (*p* < 0.05). As shown Figure 8B, no apparent difference in CAT was observed between mice receiving CZON and the control animals. As shown in Figure 8C, the MDA content of the CZON groups (35 and 50 mg/kg) increased beyond than that of the control group (*p* < 0.05 or *p* < 0.01). No obvious difference was observed of GSH between the control animals and the mice received CZON, except the 50 mg/kg group (*p* < 0.05) (Figure 8D).

### 3.6. Electron Paramagnetic Resonance (EPR) Result of CZON

Hydroxyl radicals produced by CZNO were measured using electron paramagnetic resonance (EPR). As shown in Figure 9A,B, after CZNO, suspensions were incubated with DMPO and subjected to 5 min of UV light irradiation, samples exhibited EPR spectra as well as a prominent 1:2:2:1 quartet EPR spectra. This confirmed the existence of the DMPO–OH adduct, which has a split center at 3400 Gauss, as well as the formation of hydroxyl radicals.

### 3.7. TEM and WB Results

The double membrane structure vesicle (Figure 9C–E) indicated that autolysosomes were found in the hepatocyte of mice which received CZON at 50 mg/kg by i.v. These results suggested that CZON induced the autophagy of the liver when they entered the blood by injection. WB results of LC-3II and p62 in the liver confirmed the autophagy hypothesis in the liver after administration of CZON. As shown in Figure 9F–H, compared with the control group, the LC-3II and p62 levels of mice which received CZON at different dosages increased remarkably. A lot of literature suggested that oxidative stress is an important inducement of autophagy [40,41,42]. In the present study, we found that CZON caused the oxidative stress, such as variation of SOD, MDA, CAT in major organs. Furthermore, we confirmed the presence of hydroxyl radicals produced by CZNO using EPR. Taking all of these results into account, we raised the assumption that CZON induced the autophagy in the liver by oxidative stress, especially hydroxyl radicals.

### 3.8. Cu Content Analysis

As illustrated in Figure 10, the highest Cu levels were detected in the liver compared to other organs. The mobilized peak of Cu in the liver and spleen appeared at 24 h, then the Cu level declined over the next 4 weeks. Even if the increase in Cu in other organs at 24 h after the injection was determined, this increase was relatively smaller than the liver and spleen tissues. According to the gradual degradation of Cu levels, it required more than 4 weeks for CZON to be finally excreted from the body and cleared entirely.

## 4. Discussion

As a new type of nanomaterial, ternary materials, such as CZON, i.e., Cu-doped-ZnO, have attracted more and more attention in recent years due to its unique physical and chemical properties, which exhibited excellent potential in the field of catalysis, medicine, and environment science [43,44,45]. However, there is still very little research on their toxicity. Siddique et al. evaluated the toxicity using drosophila melanogaster [46]. Recently, Fkiri et al. reported the low aquatic ecotoxicity of Cu-doped ZnO nanoparticles [45]. However, most researches focused on the toxicity of NMs in vitro. In the present study, the toxicity of CZON in vivo was evaluated systematically. It was interesting to find that CZON could induce hepatocyte autophagy when they entered the blood via the i.v. route.

Autophagy is a conserved catabolic process involving large protein degradation by a ubiquitous autophagosomic signalling pathway, which is essential for cellular homeostasis [47]. Many factors can induce autophagy, including starvation, stress, pathogeny, and so on. It is well known that oxidative stress plays an important role in the autophagy. The DAS group proved that copper doping in ZnO nanoparticles induced cytotoxicity in macrophages by ROS induction and apoptosis [48]. The formation of oxidative stress was because of the imbalance between the antioxidant defensive system and the production of reactive oxygen species (ROS), which were recognized as deleterious substances due to their hazard to cell membranes and DNA [49,50] Oxidative damage possesses the ability to cause cell damage as well as cell death and then, in turn, exacerbate chronic diseases such as cancer, fatty liver, hepatic fibrosis, and Alzheimer’s disease. In recent decades, oxidative stress caused by NMs has garnered increasing attention [51,52,53]. In particular, the role that oxidative stress plays in the organs damage caused by NMs has aroused considerate concerns by many researchers [54,55,56]. In the present study, CZON was found to potentially cause the oxidative stress to the liver and lung tissues, indicating that the liver and lung might be the major target organs of CZON after intravenous injection. Many researches have confirmed that the liver and spleen were the major target organs of inorganic nanomaterials [57]. The present study not only validated that the liver was the target organ of CZON, but also made further hepatocyte autophagy caused by CZON using a TEM observation of the autophagic vacuole. Although nanoparticles could enhance the autophagy effect in the liver cell lines, the mechanism was needed in further researches [58].

## 5. Conclusions

In conclusion, the acute toxicity and biodistribution of CZON after orally administrated was investigated. CZON appeared in a dose-dependent manner in terms of body weight, severity of pathological damage, and hematologic and blood biochemical parameters after single injection. These results indicated that CZON is able to induce oxidative damage to the liver and lung by producing hydroxyl radicals at a higher dose. Furthermore, autophagy induced by CZON by oxidative stress was supposed to be the mechanism of toxicity of CZON. However, further evaluation of the excretion is still demanded, and the future studies based on these data will provide many useful references for the development of CZON in biomedical application.

## Figures and Tables

**Figure 1 nanomaterials-11-02081-f001:**
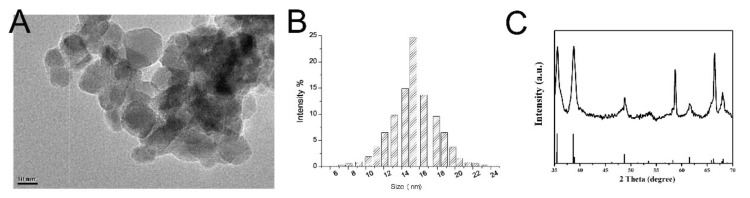
(**A**) the TEM images and (**B**) the average size of CZON by Zetasizer 3000HSA. (**C**) XRD spectra indicate that the prepared nanocrystals is CZON.

**Figure 2 nanomaterials-11-02081-f002:**
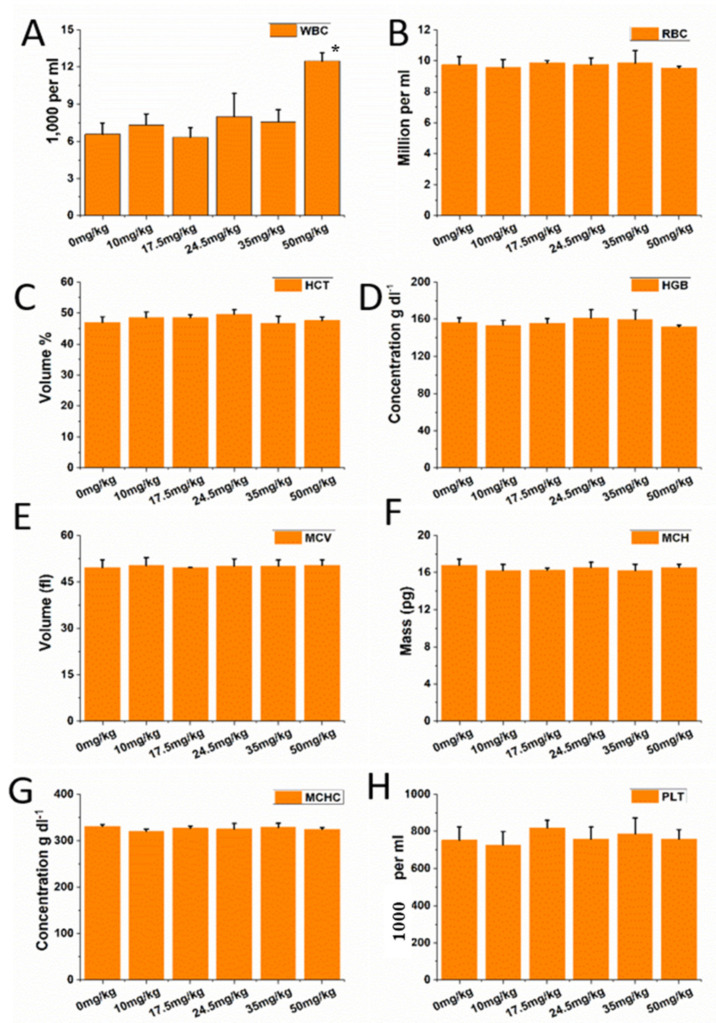
Complete blood counts of ICR mice following injection of CZON. (**A**) Mean and standard deviation of red blood cell numbers. (**B**–**D**) Mean and standard deviation of hemoglobin concentration, hematocrit (**E**)mean corpuscular volume (MCV), (**F**) mean corpuscular hemoglobin (MCH), (**G**,**H**) mean corpuscular hemoglobin concentration (MCHC), white blood cells, or platelet of ICR mice (*n* = 10). (* denotes statistical significance for the comparison of control, * *p* < 0.05).

**Figure 3 nanomaterials-11-02081-f003:**
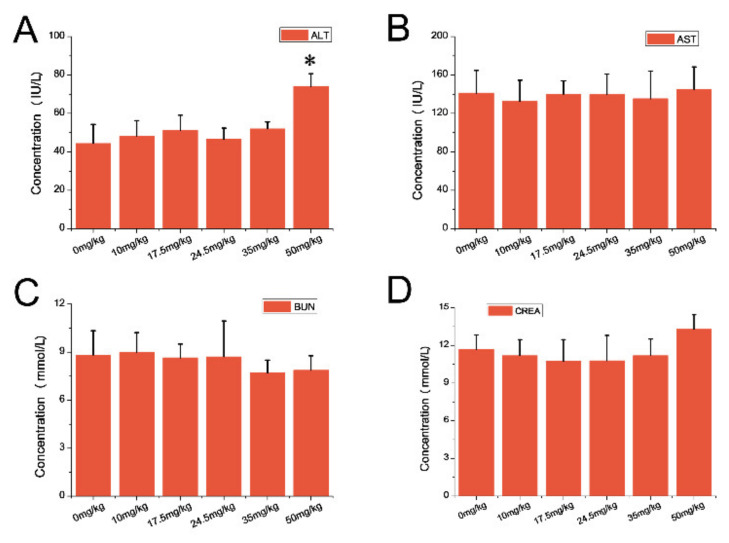
Biochemistry indexes of ICR mice following single injection of CZON. (**A**) Mean and standard deviation of alanine aminotransferase (ALT); (**B**) Mean and standard deviation of aspartate aminotransferase (AST); (**C**) Mean and standard deviation of blood urea nitrogen (BUN); (**D**) Mean and standard deviation of creatinine (CREA) of ICR mice (*n* = 10 per group). The serum ALT level of the 50 mg/kg groups increased significantly (*p* < 0.05), compared to with the control group. (* denotes statistical significance for the comparison of control, * *p* < 0.05).

**Figure 4 nanomaterials-11-02081-f004:**
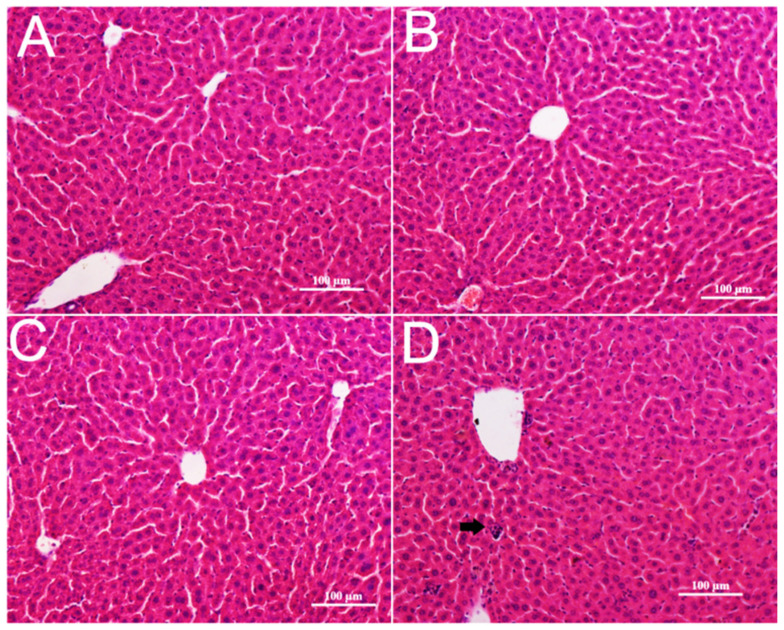
Hematoxylin and eosin stained images of the liver from control (**A**) and mice injected CZON at the dose 24.5 mg/kg (**B**), 35 mg/kg (**C**) and 50 mg/kg (**D**). Black arrow indicates lymphocytic infiltration. (Bar is 100 μm).

**Figure 5 nanomaterials-11-02081-f005:**
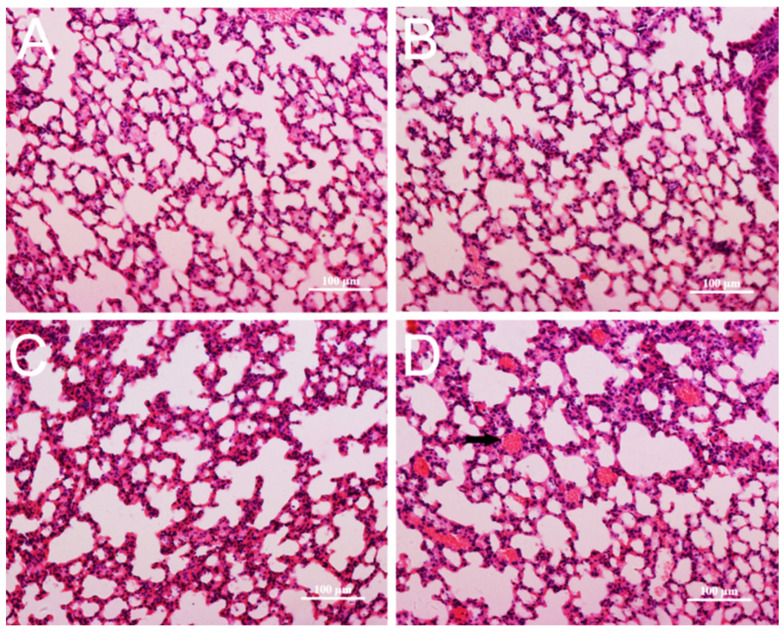
Hematoxylin and eosin stained images of the lung from control (**A**) and mice injected CZON at the dose 24.5 mg/kg (**B**), 35 mg/kg (**C**) and 50 mg/kg (**D**). (Bar is 100 μm).

**Figure 6 nanomaterials-11-02081-f006:**
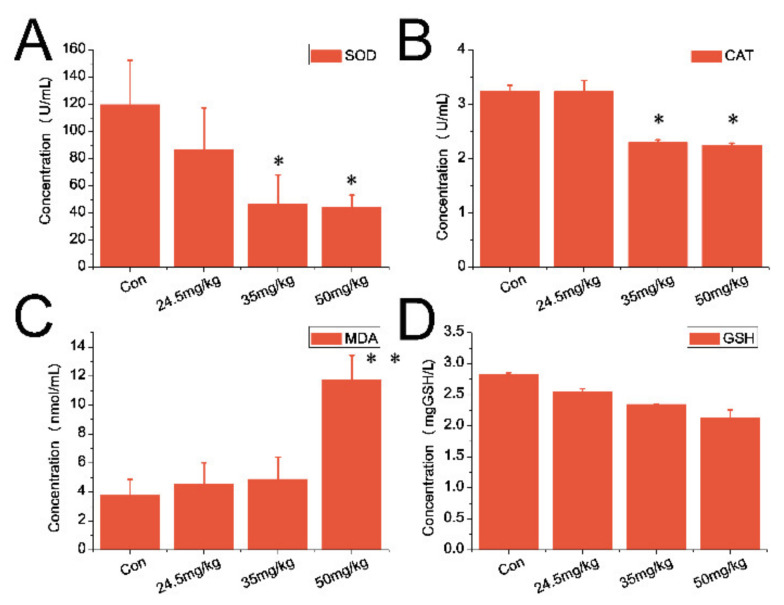
Oxidative stress indexes in the serum of ICR mice following a single injection of CZON. (**A**) Mean and standard deviation of SOD; (**B**) Mean and standard deviation of CAT; (**C**) Mean and standard deviation of MDA; (**D**) Mean and standard deviation of GSH of ICR mice (*n* = 5 per group). (* denotes statistical significance for the comparison of control, * *p* < 0.05, ** *p* < 0.01).

**Figure 7 nanomaterials-11-02081-f007:**
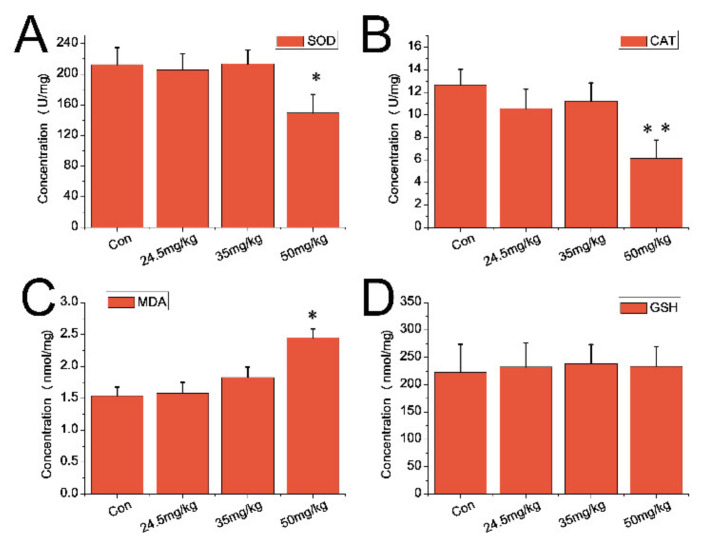
Oxidative stress indexes in the liver of ICR mice following a single injection of CZON. (**A**) Mean and standard deviation of SOD (**B**) Mean and standard deviation of CAT; (**C**) Mean and standard deviation of MDA; (**D**) Mean and standard deviation of GSH of ICR mice (*n* = 5 per group). (* denotes statistical significance for the comparison of control, * *p* < 0.05, ** *p* < 0.01).

**Figure 8 nanomaterials-11-02081-f008:**
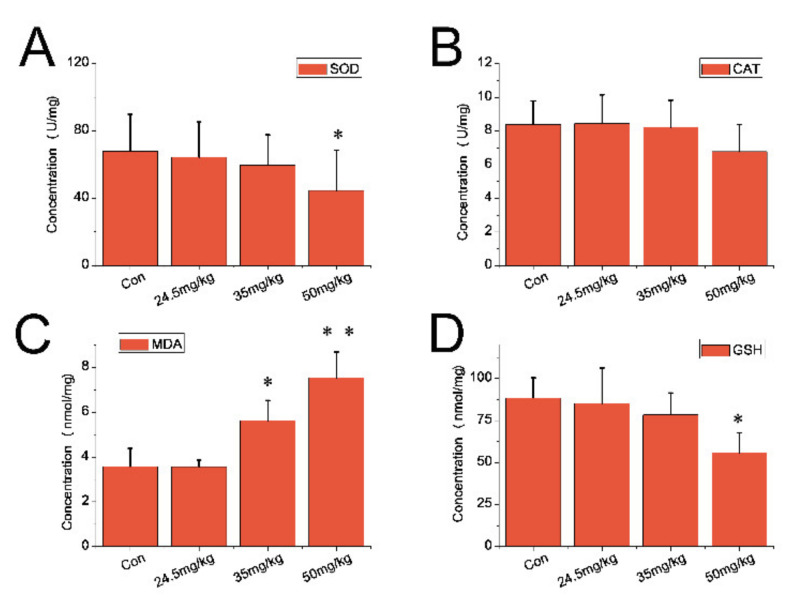
Oxidative stress indexes in the lung of ICR mice following a single injection of CZON. (**A**) Mean and standard deviation of SOD; (**B**) Mean and standard deviation of CAT; (**C**) Mean and standard deviation of MDA; (**D**) Mean and standard deviation of GSH of ICR mice (*n* = 5 per group). (* denotes statistical significance for the comparison of control, * *p* < 0.05, ** *p* < 0.01).

**Figure 9 nanomaterials-11-02081-f009:**
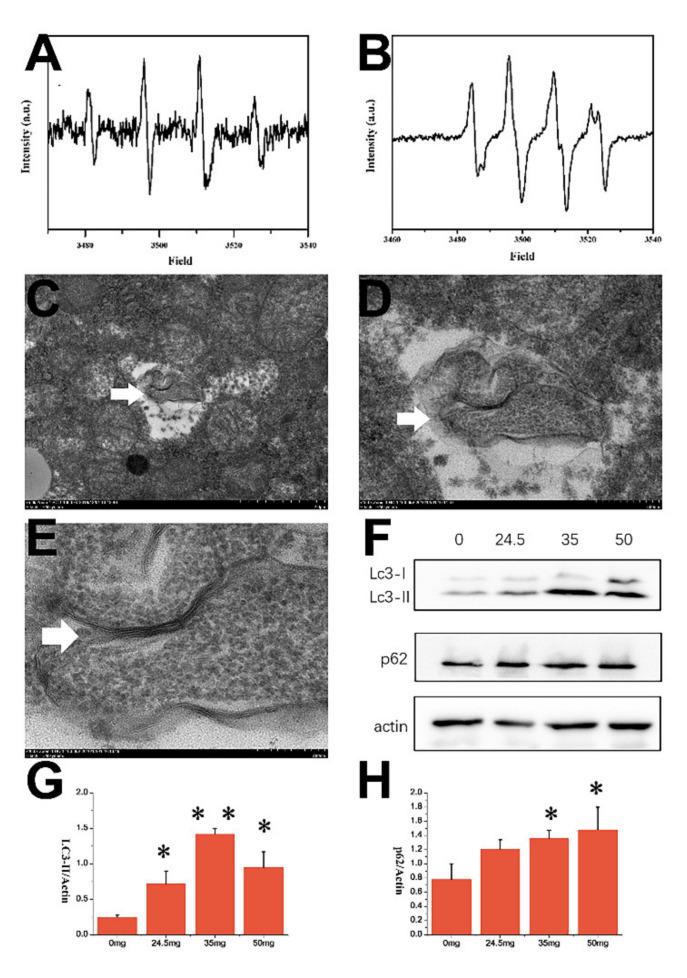
The EPR spectra of hydroxyl radicals produced of CZON (**A**) and control (**B**). TEM images of autolysosome in the liver ((**C**) 5 k, (**D**) 15 k, (**E**) 40 k). The white arrow shows the autolysosome. (**F**) Western blot result of LC-3 and p62. (**G**) LC3-II/actin results of different groups, (**H**) p62/actin results of different groups (* denotes statistical significance for the comparison of control, * *p* < 0.05).

**Figure 10 nanomaterials-11-02081-f010:**
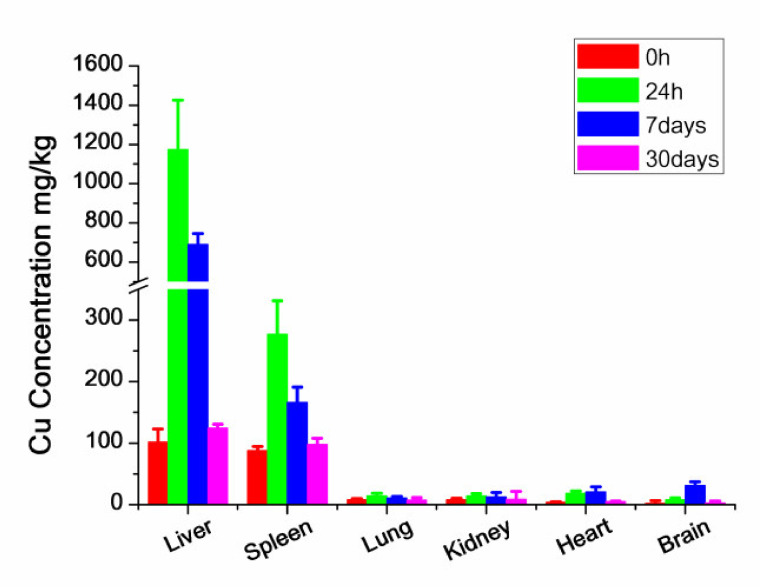
ICP-OES analysis result of Cu levels in the liver, spleen, lung, kidney, heart and brain of animals treated with CZON (*n* = 5 per group).

## Data Availability

Not applicable.

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
