# Peer review of "Cu-Doped-ZnO Nanocrystals Induce Hepatocyte Autophagy by Oxidative Stress Pathway"

_nanomaterials, 2021, doi:10.3390/nano11082081_

Round 1
Reviewer 1 Report
The manuscript entitled “Cu-doped-ZnO Nanocrystals Induce Hepatocyte Autophagy by Oxidative Stress Pathway” constitutes a preliminary evaluation of the acute systemic toxicity and biodistribution of Cu-doped-ZnO nano-11 crystals (CZON) in an animal model. CZON have emerged in recent years as potential therapeutic candidates due to the promising antifungal and anti-neoplastic properties displayed. Therefore, in view of the potential application of CZON in medical applications, the in vivo evaluation of the systemic distribution and toxicity these nanomaterials constitute a relevant pursuit.
The study is well designed and methodologies used are adequate, shading some light into the main target organs for the toxicity of CZON and for the role of CZON-induced oxidative stress in the onset of the toxic effects produced by these materials. Therefore, recommend this paper for publication.
Minor issue:
Page 2 line 49: please rephrase the sentence: “Especially in anti- bacterial, CZON have been the novel antibacterial agents due to the insecure surface chemical properties.”, CZON are not novel antibacterial agents, they showed interesting properties in this field and can be regarded and potential new antibacterial agents
Author Response
Referee 1:
Please rephrase the sentence: “Especially in anti-bacterial, CZON have been the novel antibacterial agents due to the insecure surface chemical properties.”, CZON are not novel antibacterial agents, they showed interesting properties in this field and can be regarded and potential new antibacterial agents.
Response:
Thanks for your suggestion. We have corrected page 2 line 52 to “CZON have been the promising new antibacterial agents due to the insecure surface chemical properties.” We are very sorry for the line number of the page may have changed because we added some content to make the manuscript more complete.
Reviewer 2 Report
The authors in this manuscript study the in vivo toxicity (in rats) of the new nanoparticles CZON. These studies have a high degree of innovation since these nanoparticles have so far only been tested on cell cultures.
Strength:
Detailed biochemical and toxicological (histological) analysis of CZON nanoparticle effects in rats (first performed experiments)
Weakness:
The physicochemical characteristics of CZON nanoparticles could have been presented a little better.
Minor:
- In the introductory part of the manuscript, the following terms should be explained in one sentence: nanomaterials (size range), autophagy
- It is necessary to give full names in abbreviations in the introductory part, since they are mentioned here first: MDA, SOD, CAT and GSH.
- Authors in materials and methods must state the names of the following devices: DLS and Raman spectroscopy.
- The authors must state in what form the tested nanomaterials (CZON) were measured in DLS and Raman spectroscopy, whether it was a powder, suspension ....
- Which was the concentration of CZON in the suspension with glucose
Author Response
Please rephrase the sentence: “Especially in anti-bacterial, CZON have been the novel antibacterial agents due to the insecure surface chemical properties.”, CZON are not novel antibacterial agents, they showed interesting properties in this field and can be regarded and potential new antibacterial agents.
Response:
Thanks for your suggestion. We have corrected page 2 line 52 to “CZON have been the promising new antibacterial agents due to the insecure surface chemical properties.” We are very sorry for the line number of the page may have changed because we added some content to make the manuscript more complete.
Referee 2:
- Explain the following terms “nanomaterials (size range), autophagy” in one sentence.
Response:
Thanks for your suggestion. We added the explanation in the introductory portion “Nanomaterials refers to a kind of materials whose size are less than 100nm in one dimension.” on page 1 line 33 and introduced a reference, you can find it on line 370, Reference 1. We also added the explanation of “autophage” on page 2 line 61 in the introductory part as “Autophagy can cause programmed cell death by degrading membrane-isolated cytoplasm and organelles.” And Reference 25 was added, you can find it on page 13 line 431.
- Give full names in abbreviations in the introductory part, since they are mentioned here first: MDA, SOD, CAT and GSH.
Response:
We have re-written this part according to the Reviewer’s suggestion, we replaced “MDA, SOD, CAT and GSH” with “Malonaldehyde (MDA), superoxide dismutase (SOD), catalase (CAT) and glutathione (GSH)” in the introductory part.
- Authors in materials and methods must state the names of the following devices: DLS and Raman spectroscopy.
Response:
The names of devices, on line 90 and line 91, the name of these devices “DLS (Zetasizer 3000HSA. Malvern, UK), Raman spectroscopy (Raman DXR3, Thermo, USA)” were added.
- The authors must state in what form the tested nanomaterials (CZON) were measured in DLS and Raman spectroscopy, whether it was a powder, suspension.
Response:
Thanks for your suggestion. On page 2 line 91 and 92 “The size of nanomaterial was measured by DLS (Zetasizer 3000HSA. Malvern, UK) using the suspension liquid of CZON. Raman spectroscopy was also used to investigate the structure of CZON (Raman DXR3, Thermo, USA), applying CZON suspension.” was added.
- Which was the concentration of CZON in the suspension with glucose?
Response:
Thanks for your suggestion. We have rephrased the sentence in page 3 line 106 as “CZON was suspense in sterile 5% glucose (10mg/kg) and administrated through the mouse tail vein.”
Once again, thank you very much for your comments and suggestions. Looking forward to hearing from you.
Best regards!
Yours sincerely,
Tianlong Liu